# An Industrial Internet-of-Things (IIoT) Open Architecture for Information and Decision Support Systems in Scientific Field Campaigns

**DOI:** 10.3390/s24185916

**Published:** 2024-09-12

**Authors:** Yehuda Arav, Ziv Klausner, Hadas David-Sarrousi, Gadi Eidelheit, Eyal Fattal

**Affiliations:** Department of Applied Mathematics, Israel Institute for Biological Research, P.O. Box 19, Ness-Ziona 7410001, Israel; zivk@iibr.gov.il (Z.K.); hadasdavids@gmail.com (H.D.-S.); gadieide@yahoo.com (G.E.); eyalfattal@yahoo.com (E.F.)

**Keywords:** Industrial Internet of Things (IIoT), atmospheric sciences, environmental sciences, field campaigns, microservices architecture, open-source software

## Abstract

Information and decision support systems are essential to conducting scientific field campaigns in the atmospheric sciences. However, their development is costly and time-consuming since each field campaign has its own research goals, which result in using a unique set of sensors and various analysis procedures. To reduce development costs, we present a software framework that is based on the Industrial Internet of Things (IIoT) and an implementation using well-established and newly developed open-source components. This framework architecture and these components allow developers to customize the software to a campaign’s specific needs while keeping the coding to a minimum. The framework’s applicability was tested in two scientific field campaigns that dealt with questions regarding air quality by developing specialized IIoT applications for each one. Each application provided the online monitoring of the acquired data and an intuitive interface for the scientific team to perform the analysis. The framework presented in this study is sufficiently robust and adaptable to meet the diverse requirements of field campaigns.

## 1. Introduction

Field campaigns play a critical role in environmental sciences to examine micro-meteorological conditions and the dispersion of pollutants in the atmosphere or indoors [1,2,3,4,5,6,7,8]. They allow for the characterization of flow and dispersion phenomena in different scenarios and the development of mathematical models describing these phenomena [9,10,11,12,13,14,15,16,17]. The specific research goals of each campaign vary; thus, different sensor configurations are needed to gather the necessary data. Nevertheless, some similarities do exist. Usually, a field campaign comprises trials that are aimed at measuring the behavior of a system under documented conditions and controlled experimental parameters. Collaboration among multiple research teams is common due to the high cost and complexity of field campaigns, but this results in additional work to integrate and standardize the data collected by each team. This is crucial both for monitoring the field campaign during its execution, allowing for optimization between trials, and for analyzing the results after the campaign concludes.

Information and decision-support software must be flexible enough to accommodate the requirements of different field campaigns, that is, to handle different types of sensors and various analysis methodologies. However, this flexibility entails considerable costs. Developing custom software using a software framework is one possible method to reduce development costs.

Industrial Internet-of-Things (IIoT) frameworks offer services to build software that collects, analyzes, presents, and stores acquired data for future analysis. A typical IIoT application is divided into two domains: the edge and the server. The edge domain consists of the devices and instruments deployed in the field. The server domain collects sensor data, stores it in a standardized format, conducts real-time analysis, and enables user interaction. The integration of the edge and server domains is usually achieved with middleware software that is responsible for regularizing the data [18]. Physically, the sensors and actuators in the field domain are connected to the server domain directly or via a gateway. The IIoT gateway serves as a proxy for handling multiple hardware interfaces and technologies (such as WiFi, Zigbee, and a CAN-controller area network), and it also governs the persistence, provisioning, and management of devices [19,20,21,22].

Over the past two decades, IIoT frameworks have matured and been extensively used in many fields, such as logistics, retail, emergency response, smart cities, and smart factories [23,24,25,26].

IIoT frameworks have been developed by commercial companies, including Microsoft Azure IoT [27], Amazon Web Services (AWS) IoT [28], Google Cloud IoT [29], IBM Watson IoT [30], Siemens MindSphere [31], GE Digital Predix [32], PTC ThingWorx [33], Bosch IoT Suite [34], SAP Leonardo IoT [35], and Honeywell Forge [36]. The open-source community has also developed numerous frameworks, including Eclipse Kura [37], IoTivity [38], OpenHAB [39], EdgeX Foundry [40], OpenIoT [41], Apache Edgent [42], and Eclipse Ditto [43]. Commercially available tools provide a comfortable and complete environment for developing an IIoT application. However, they are currently expensive and usually cloud-based, which could limit their use in field campaigns where internet broadband is limited. Open-source tools provide different costless components and support deployment on local servers (on-premise), but they require additional effort to integrate them into a single application.

Open-source IIoT frameworks, and IIoT frameworks utilizing open-source components, have been used in various domains, such as CNC machines [44], polymer electrolyte membrane (PEM) hydrogen generators [45], agriculture systems [46], and ferry systems [47]. However, the domain of scientific field campaigns, which is the focus of this work, differs significantly from the typical industrial use cases due to several reasons:Each field campaign is unique and requires different methods of data collection and analysis. Therefore, the solution cannot provide well-defined and fixed dashboards for real-time monitoring and analysis. Instead, it must offer a tool that allows the technical team to customize it according to the specific requirements of the field campaign with minimal effort.The real-time monitoring of data is crucial only during the field campaign. After the field campaign concludes, it is essential to provide a user-friendly interface to access the collected data and the settings (e.g., details of the trials) under which they were gathered.

The IIoT framework presented in this work functions both as an architecture and as a library for an IIoT application, supporting the rapid development of information and a decision-support system for field campaigns. A key feature of the proposed framework is that it requires minimal coding to construct a management system easily tailored to the specific needs of each campaign. Additionally, the architecture of the server domain is independent of the field domain, as it supports various protocols and can be easily adapted to different implementations of the edge domain.

The structure of the rest of the paper is as follows. Section 2 describes both an indoor and an outdoor field campaign, highlighting the differences in their requirements. Section 3 describes the framework, and Section 4 details the implementation of the framework for the two field campaigns. Finally, Section 5 discusses the strengths and novelties of the work compared to the previous literature, its main limitations, and future research directions.

## 2. Example of Indoor and Outdoor Field Campaigns

In this section, we describe indoor and outdoor field campaigns that we conducted a few years ago. While the primary motivation for these campaigns was scientific research in the field of air pollution dispersion, the results provided valuable insights for transit planners and civil engineers. Specifically, they highlight the impact of mass transportation systems and industrial facilities on air quality within train stations and urban streets. The measuring instruments used in these campaigns were similar to those employed in air quality surveillance [48]. Such surveillance is typically conducted to monitor emissions from major industrial plants and highways, as required by municipal and governmental regulations [49,50].

This is done to highlight and illustrate the variety of field campaigns and the corresponding requirements from the IIoT framework. We will only detail the technical aspects of the field campaign that are necessary to understand the function of the IIoT framework.

### 2.1. Indoor Field Campaign

This field campaign was conducted to study tracer gas dispersion in an underground train station, and it took place at the Allenby train station in Tel-Aviv between 7 and 14 March 2022. During the campaign, $SF_{6}$ was released as a tracer gas from different locations in the station (Figure 1A). Each release was conducted at a different time, under various conditions, and named using a combination of the release point and a sequential number. Table 1 lists a selection of the trials in the campaign.

In the campaign, the concentrations of $SF_{6}$ were measured via 65 nondispersive infrared (NDIR) sensors [51] at a frequency of 1 Hz, and their locations for trials A1–A4 and B1–B4 are depicted in Figure 1B. The micro-climate of the station was measured using 45 temperature sensors that sampled at a frequency of 1 Hz (PT1000, [52]) and 3 Kaijo ultrasonic wind anemometers (DA-650) with a TR-92T probe that sampled at a 20-Hz frequency [53]. The Kaijo anemometers were positioned in different locations in trials A1–A3, A4, and B1–B5 to sample the microclimate in the station (Figure 2). The data from all these instruments were collected using a datalogger [54].

The data collected via the different devices were transmitted online to the server side using the UDP protocol [55]. Each sampling point was transmitted as a string with the following format:[Device Name], [Device number], [Time stamp], [device specific fields], XXX

For example, a typical message from the wind anemometer sensor is
$$\mathrm{Sonic}001,2022/8/1\;10:30,0.05,\underset{U,V,W,T}{\underset{︸}{-0.3,0.03,0.02,23.2}},\mathrm{XXX}$$
where U, V, W, and T are the measured wind speed in the x, y, and z directions (respectively), and T is the temperature.

The technical team required the real-time monitoring of the frequency of messages from each device, while the scientific team required the real-time analysis of the covariants, computed in segments of 1, 5, and 10 min, of the micrometeorological data. The covariants for a window of *X* minutes were computed by dividing the data into *X*-minute segments and computing the mean values of each batch (for example, $\bar{u}$ and $\bar{v}$ represent the means of the *u* and *v* components). Then, the mean values were subtracted from the raw data to obtain the fluctuations (i.e., $u^{\prime }=u-\bar{u}$ and $v^{\prime }=v-\bar{v}$). Finally, the fluctuations were multiplied and averaged over the *X*-minute window to compute the different covariants (for example, $\bar{u^{\prime }v^{\prime }}$). Additionally, rapid access to the concentration data facilitated the ability of the research team to select the optimal conditions under which the next trial would be conducted.

### 2.2. Outdoor Field Campaign

This field campaign was conducted to study the ground deposition of droplets in urban settings. It took place at the Urban Warfare Training Center in Israel between 24 September 2023, and 5 October 2023. During the campaign, Propylene Glycol (CAS 57-55-6) that contained a food colorant additive was dispersed at a height of 10 m and collected using 300 Petri dishes and sampling sheets of paper placed on the ground (Figure 3). After each trial, the Petri dishes and the sheets of paper were collected and analyzed in a field lab. The quantity of the liquid collected in the Petri dishes was sampled to analyze the amount deposited, and the sheets of paper were photographed and then analyzed using an image-processing technique. This technique was used in order to estimate the sizes of the particles that created the color stains collected on each paper. Hence, unlike in the indoor field campaign, the data had to be analyzed in a lab and reported to the scientific team a few hours after it was collected.

The vertical wind and temperature profile was measured at 3 heights using three-dimensional ultrasonic anemometers [56] and temperature and humidity sensors [57] that were stationed on a 20-m-high mast. The sampling frequency of the ultrasonic anemometers was 32 Hz, and that of the temperature and humidity sensors was 1 Hz. The data from all these instruments were collected via a datalogger [54]. Alongside these instruments, solar radiation and far infrared radiation were measured using a net radiometer device [58]. The net radiometer device was posted upon a 1.8-m tripod located alongside a 20-m meteorological mast. The sampling frequency of the net radiometer was 1 Hz, and the data were sent to the datalogger [54] that collected the devices’ inputs. The dataloggers’ memory served as a preliminary data backup. The data were also transmitted via a cellular modem [59] to the main data server.

Within the streets, meteorological measurements were conducted using 3 portable meteorological stations that each consisted of three-dimensional ultrasonic anemometers [56] and a temperature and humidity sensor [57]. Two portable stations were erected on a tripod up to a height of 3 m, and the third station was erected on a pneumatic mast that was raised to a height of 10 m. The data collected from each of the stations were sent via a cellular modem [59] to the main data server.

## 3. A Framework for an Information and Decision-Support System for Field Campaigns

Field campaigns are generally conducted in three phases: planning, execution, and post-analysis. During the planning phase, the research team determines which sensors will be used, and their location in the field determines the controllable environmental aspects in each trial, and the team designs real-time, on-screen dashboards and analysis procedures to show the campaign’s status. In the execution phase, the research team deploys the experimental array and monitors the data acquired to ensure proper operation. During the execution phase, the research team analyzes the data obtained and determines whether any adjustments to the experimental array are necessary. In the post-analysis phase, which takes place after the campaign, the research team performs an in-depth analysis of the data offline.

The framework presented in this work consists of three modules that handle the planning, execution, and post-analysis phases. Each module consists of one or more open-source components. We will describe the different components and architectures of each module in the following sections.

### 3.1. Planning Module

In the planning phase, the scientific and technical teams determine which trials will be conducted, as well as which sensors will be deployed and where. Hence, it is desirable to have a standard methodology to record the details of the trials. However, the authors are unaware of any open-source package that could be used to plan a campaign. Therefore, we have developed ArgosWeb [60], an open-source, web-based application that allows the user to set up a field campaign. We will briefly describe the ArgosWeb software; for a thorough explanation of the different features, see the built-in documentation.

Figure 4 depicts the ArgosWeb interface. First, the user defines a new field campaign and determines its name (Figure 4, box 1). Then, the user defines the properties that define each trial. Since each group of trials is designed to investigate different aspects of a natural phenomenon, the properties that define each trial differ between field campaigns. For example, each trial in the indoor field campaign (Table 1) was described using the following properties: start time, end time, release point, amount released, release time, and the state of the entrance doors as open or closed (Figure 4, box 2). Once the properties are defined, the user can fill in the details of each trial.

After the trials were defined, the user could then define the devices in the system. To keep a complete record of the settings, each device is described according to its physical location and other properties that can be defined for each device type (Figure 4, box 3). For example, the sampling height of the NDIR sensors was recorded. The value of these properties might depend on the trial. Setting the values of the devices in each trial is shown in Figure 5.

Finally, some field campaigns require the uploading of images that could either be embedded on a map itself (for example, to increase the resolution) or be used as a map (Figure 4, box 4). The indoor field campaign, for example, required the schematics of the concourse and the platform of Allenby Station.

After the trials and the devices are defined, the scientific and technical teams can plan the deployment of the sensors on the map in each trial and set the value of the trial-dependent properties (Figure 5). In Figure 5, we show the example of determining the deployment of the NDIR devices on the concourse floor of the Allenby subway station. In the outdoor field campaign, the deployment took place in a field campaign.

After the field campaign is defined, it is possible to download the description as a ZIP file that contains the definitions of the trials and devices, as well as all the images. The pyArgos library was developed to provide a Python wrapper for the ZIP file [61]. Using the wrapper, it is possible to get the list of trials and the devices with the value of the different properties in each trial as a pandas data frame [62]. This allows the user to query and plot the schemes required for the technical team during the deployment in the field. The pyArgos code is available in GitHub [61].

### 3.2. Execution Module

The execution module is responsible for acquiring the data from the sensors, storing it in a standard format, performing the real-time analysis (e.g., computing statistics over different time windows), and displaying the campaign’s status to the research team. Since field campaigns employ various sensors, each campaign requires custom analysis procedures. Hence, it is necessary to ensure that it is easy to adapt the execution module to the campaign’s specific needs.

This stage of real-time analysis has great advantages in allowing the research team conducting the campaign to identify ongoing problems in the measurements (incorrect values or technical malfunction). In addition, real-time analysis allows the research team to validate some of their planning hypotheses even during the field campaign. The manager can adapt the deployment of the sensors in the field if a planning hypothesis is invalid.

In the following subsection, we describe the general architecture of the server domain (Section 3.2.1), and then we describe how to implement it using open-source components (Section 3.2.2).

#### 3.2.1. The Architecture of the Server Domain

The architecture of the application in the server domain comprises 5 functional layers (Figure 6), that were defined in previous IOT workgroups [63,64].

The normalization and connectivity layer (Figure 6, blue box), also known as the middleware layer, receives the data from the sensors in the field and their status.

The data received are transformed into a standard format and stored via the data management layer (Figure 6, yellow box). The device management layer manages the status of the device (Figure 6, green box). Specifically, it ensures that the device is online, records errors, and is responsible for updating the software on edge devices.

The event and analytic layer perform the statistical analysis required by the research team (Figure 6, orange box). The calculation is performed either on demand by the user or periodically.

The application layer enables access to the data and status of the devices via a graphical user interface (Figure 6, purple box).

#### 3.2.2. Implementation Using Open-Source Components

Implementing the architecture depicted in Figure 6 requires specifying each layer’s application programmable interface (API). One way to decrease the workload and accelerate development is to use open-source software components. An open-source component is computer software released under licensing that permits usage, distribution, and modification [65]. Since open-source components are maintained by the community and free of charge, they provide a cheap and reliable way to accelerate the development of applications.

The abundance of IIoT applications in various industry domains increased the interest in open-source solutions [66]. However, the authors are unaware of a single open-source component that fulfills the requirements for managing field campaigns. Therefore, the approach we chose was to integrate several open-source packages. The challenge in this approach is to develop a single application that integrates the different open-source components. The integration is particularly challenging because each field campaign has specific requirements and might include in-house analysis code from different research groups.

To simplify the integration, we used the microservices pattern to reduce integration costs while maintaining flexibility [67]. In this pattern, each layer is structured as a collection of independent, loosely coupled components that receive requests and produce output. Because the components are independent, they can be added, removed, or upgraded without interfering with any other application function. Hence, the question of integration reduces to finding a methodology to transfer messages between the different services that support different protocols. Using the microservices pattern requires adding the messaging layer to the architecture. This layer is responsible for conveying messages between the various components.

Our implementation of the architecture of Figure 6 as a microservices pattern with open-source components is depicted in Figure 7. In the following, we delineate the implementation of each layer with open-source components.

The messaging layer is responsible for connecting the different microservices, and it is implemented with the Apache Kafka [68] and Node-RED [69] packages.Apache Kafka is a messaging service that delivers key–value messages to an arbitrary number of consumer processes [68], and as such, it constitutes the main backbone of the messaging system (Figure 7). One advantage of Apache Kafka is the fail-safe mechanism that handles failures in the server or in any consumer. To handle failures in the server, Kafka can be installed in a cluster, which prevents data losses in the case of a server failure [68]. To handle failures in the consumer, the server maintains a record of the messages that were successfully delivered, and it can fill in missing messages at later times. Another advantage of Apache Kafka is that its interface is implemented in many programming languages (more than 19 currently), and therefore, it simplifies the integration between different research teams that developed their codes in different programming languages.However, using the Apache Kafka interface requires integration with the code, which might require considerable effort or even be impossible for a commercial third-party code. To solve this problem, we use the Node-RED package [69] as an adaptor between the different components and the Apache Kafka messaging services (Figure 7).Node-RED [69] is a graphical flow-based programming tool for building extract–transform–load (ETL) applications. Each data item is represented as a message (a collection of key–values) that is transformed with each stage (node) of the workflow. The user connects nodes that perform a specific action to construct a workflow. Node-RED has a large repository of nodes that transform the content of the message, handle input and output, and determine the logic of the flow. In our framework, a designated workflow serves as the adaptor between Kafka messages and the specific protocols of the different components (see the normalization and connectivity layer below for an example). Since the programming of a workflow is done using a drag-and-drop interface for the different nodes, it is highly flexible and enables the user to adapt the workflow to specific needs with little effort. Node-RED has over 3000 nodes in its repository, and it is relatively simple to develop specialized nodes using the node-js programming language [69]. We note that Node-RED is generally used as middleware that connects the field and the server domains by regularizing the data collected from the different sensors [18]. In this work, Node-RED functions as middleware in that sense, but it also serves as a component that connects different components in the server domain (e.g., the application layer).A normalization and connectivity layer transforms the data into a standard format and is implemented using the Node-RED package [69]. The specific implementation of the normalization and connectivity layer depends on the protocols used by the devices in the field domain. For many communication protocols (such as TCP, UDP, CoAP, Modbus, CANBus, or reading from a file) it is sufficient to use only Node-RED, but some protocols require a specialized component. For example, the Mosquitto package [70] is required for the MQTT protocol, which is considered a standard for IoT ecosystems [71].In the field campaign presented in Section 2, the devices send the data in the UDP protocol using the format detailed in Section 2.1. The processing of each message is implemented as a Node-RED workflow (Figure 8). The ’UDP’ node (Figure 8, gray node) receives the message from the designated port in the UDP protocol. Then, the ’find device type’ node (Figure 8, light orange node) adds the device type to the message. The ’router’ node routes the message (Figure 8, yellow node) to the parsing nodes according to the device type (Figure 8, dark orange node). The message parsing consists of transforming it from a list of values separated with commas (the format in Section 2.1) to key–value pairs, where the key is the field’s name and the value is the actual data that was transmitted. Hence, the keys in each message after parsing depend on the type of device. Finally, the parsed message is transmitted to the Kafka messaging service (Figure 8, white node).The data management layer stores the data received from the normalization and connectivity layer. Since each device sends different information, we used MongoDB [72], a noSQL database that stores data as a key–value document, and therefore, it does not require any prior information on the structure of each message. We used a Node-RED workflow to store the Kafka data message sent via the normalization and connectivity layer as a record in MongoDB (Figure 7).The device management layer manages the state of the device. In this framework, we used the ThingsBoard package [73] to keep track of the device status. Thingsboard is an open-source IIoT platform that enables the rapid development, management, and scaling of IIoT applications. The package lets the user track the device’s state, build rules that trigger alarms, and design and present a real-time dashboard with the simple dragging and dropping of different graphical components (widgets). Thingsboard has a rich library of widgets for the presentation of pie charts, line charts, histograms, gauges, tables, control buttons, and map widgets. It is also possible to design and develop new widgets using Hypertext Markup Language version 5 (HTML5) and JavaScript.In the analytical layer, the data analysis is performed using dedicated routines that are usually programmed in Python or Java. Typical analysis routines receive a computational request via Kafka, query the data from MongoDB, perform a calculation, and send the result back. The computation request can be sent either from the GUI or periodically from a dedicated Node-RED workflow.Through the application (user interface), the research team can access the data acquired via the sensors and computed through the analytical layer via real-time dashboards or interactive workbooks.Real-time dashboards show the data online. As the analytical layer description mentioned, the ThingsBoard package [73] enables the user to build a dashboard using a simple drag-and-drop interface. The dashboard is published as a web page that can be accessed by other users.

An interactive workbook is a complementary methodology to present and analyze the data. Workbooks allow the research team to perform complex analysis procedures in various programming languages (Python, R, Scala, Julia, etc.). The Jupyter package [74] allows the user to build a workbook that consists of formatted text, code, and results. The workbook results can be exported to an HTML page or portable document format (PDF). The interactive workbook allows the research team to analyze the data acquired during the field campaign.

## 4. The IIoT Application in the Indoor and Outdoor Field Campaigns

In this section, we describe the IIoT application that was developed for the scientific environmental field campaigns described above (Section 2).

### 4.1. Indoor Field Campaign

The data flow in the indoor field campaign is depicted in Figure 9. The IIoT application was designed to provide two types of analysis: technical analysis every 10 s and physics-related analysis every 30 s. The technical analysis computed the number of messages from each sensor to ensure that the device was active. The physics-related analysis computed first- (e.g., means, quantiles, etc.) and second-order (e.g., correlations of the fluctuations) statistics in windows that consisted of the data of the last 1 to 15 min. In addition, the IIoT application also provided offline access to all the data that were collected in the trials, using the same computational code in Jupyter [74].

The architecture of the IIoT application was as follows. The data collected from the devices were transformed from a string with comma-separated values (CSV)-based data into a Javascript object (JSON) format using Node-RED. Then, the record was stored in MongoDB. Each device type was stored in a different collection.

The technical and physics-related analyses were initiated periodically via a specialized Node-RED workflow, every 10 and 30 s, respectively (Figure 9, arrow 1). The workflow sent a computation command to the Python client via the Kafka server (Figure 9, arrow 2). Once the client received the message, it queried the MongoDB, computed the required statistics, and published the results to a different topic in Kafka (Figure 9, arrow 3). Another designated workflow received the statistics and published them to Thingsboard for online monitoring in dashboards (Figure 10).

It is important to note that the code libraries used to compute the statistics were also available to the offline component. Hence, the scientific team could produce ad hoc computations during the field campaign. The IIoT application in this campaign was hosted on a Cisco C240-M5 server, with two Intel Xeon 6150 CPUs and 384 GB of RAM (purchased from Bynet communication, Tel Aviv-Yafo, Israel).

### 4.2. Outdoor Field Campaign

The data flow in the outdoor field campaign is depicted in Figure 11. The meteorological data (sonic anemometer, temperature, humidity, and radiation) are transmitted as a comma-separated values (CSV)-based string, transformed with Node-RED to a JSON record, and sent to the Kafka server. The Petri dishes and the sheets of paper were collected from the field after the trial and analyzed in the laboratory. The output of the laboratory was text files in the CSV format that were loaded and converted into JSON using Node-RED.

A Python client was activated every 5 min, read all the messages accumulated since the last activation for each device type, and appended the data to an Apache parquet file. Apache Parquet is a free and open-source, column-oriented data storage format in the Apache Hadoop ecosystem, and it is supported by the Pandas data analysis library in Python.

The data saved as a Parquet format were available to the technical and scientific teams using Jupyter. Similar to the indoor field campaign, the descriptions of the trials and devices defined using the ArgosWeb interface were accessed through the pyArgos python wrapper. The IIoT application in this campaign was hosted on a Lenovo P720 Tower workstation with a single Intel Xeon 4214 CPU and 128 GB RAM (purchased from Y.A. Mittwoch & Sons, Tel Aviv-Yafo, Israel).

## 5. Discussion

This work introduced a framework for developing the server side of information and decision-support systems’ applications for field campaigns. Using this framework simplifies and accelerates the development of software tailored to the specific requirements of the field campaign, in particular to the data protocols of different devices, analysis procedures, and presentations. Due to the flexibility of the design, the server side is independent of the implementation of the edge domain.

The framework uses the micro-service pattern [67], which arranges the application as a collection of loosely coupled components (or services), for example, a component that stores the raw data in the database or another that calculates the first and second moments of sonic anemometer data. Since different micro-services are isolated, this pattern simplifies the addition, removal, or upgrading of the functionality required for the field campaign.

Simple campaigns that employ up to a few hundred devices could be managed from a laptop or a desktop. In campaigns that employ a higher number of sensors, it is possible to use servers or distribute the different micro-services among several machines.

The different services were implemented, when possible, using popular and well-supported open-source IIoT components. Using open-source components is beneficial because it reduces costs and decreases the dependence on a specific component implementation (e.g., the vendor-locking problem). Since IIoT is beneficial in many fields, many different IIoT open-source packages are available. Consequently, many different combinations of different packages could lead to very similar results. In this work, we have presented a framework consisting of a certain set of IIoT packages and the methodology to integrate them with minimal coding effort. Based on our experience, this resulted in a decrease in the costs and time required to develop an IIoT application for a specific field campaign.

The applications that were developed using the infrastructure described above have effectively implemented the open architecture paradigm. It is simple to extend the application’s capabilities by adding, upgrading, or swapping the components of the system beyond its original capabilities. This is because combining the micro-service pattern and Kafka and Node-RED components as the integration infrastructure allows the redirection of messages between the different components with a minimal amount of coding and modification of configuration files. Consequently, it is possible to integrate simplified predictive models that assimilate online measurements in order to produce real-time forecasts for the benefit of the research team. Also, since using sensors on drones has become more common, it allows for implementing a closed feedback loop between the fixed sensors in the field, computational models, and drones, making it possible to guide the drone movements according to the predictions of a mathematical model. Hence, when the research team conducts a similar field campaign, it is possible to adapt the existing implementation to the needs of the field campaign and decrease the effort in developing such software even further.

While we use Linux as our operating system, the solution we have presented in this work works independently of the type of operation system employed. All the needed open-source components support MS Windows, Linux, and MAC-OS or can be encapsulated using container technology such as dockers. Our implementation uses MongoDB [72] and Node-RED [69] as local services and Thingsboard as a docker service, but other combinations are possible.

## 6. Conclusions

In conclusion, we have presented a robust and adaptable framework that streamlines the development of specialized applications for data acquisition and analysis in scientific field campaigns. The framework’s applicability was demonstrated in two field campaigns with diverse requirements.

The suggested framework has the following advantages: Firstly, the microservices architecture supports the simple adaptation of the data-processing pipeline. Secondly, using standard open-source components reduces the development time without compromising system adaptability. Specifically, using Kafka as the message broker provides flexibility in selecting the computer language in which different modules are implemented; this could be an advantage when interfacing with legacy code is needed or when collaborating with different research groups. Thirdly, the developed ArgosWEB GUI provides an intuitive interface to design field campaigns.

One limitation of the present system is that incorporating a sophisticated physical analysis and integration with different computational models, such as the Weather Research and Forecasting model (WRF, [75]), requires some coding and might be a limiting factor for small academic teams. Hence, possible future research could develop libraries that integrate different standard models and provide tools to design complex analyses without coding. However, as the analysis tools are field-dependent, it would be necessary to develop the analysis tools for specific fields.

The framework presented here could also be used to develop IIoT applications in other fields, such as emergency response systems, industrial, logistics, etc. In such cases, using this framework in these fields would require taking into account other aspects, such as real-time performance and security, that are of interest in each particular field.

## Figures and Tables

**Figure 1 sensors-24-05916-f001:**
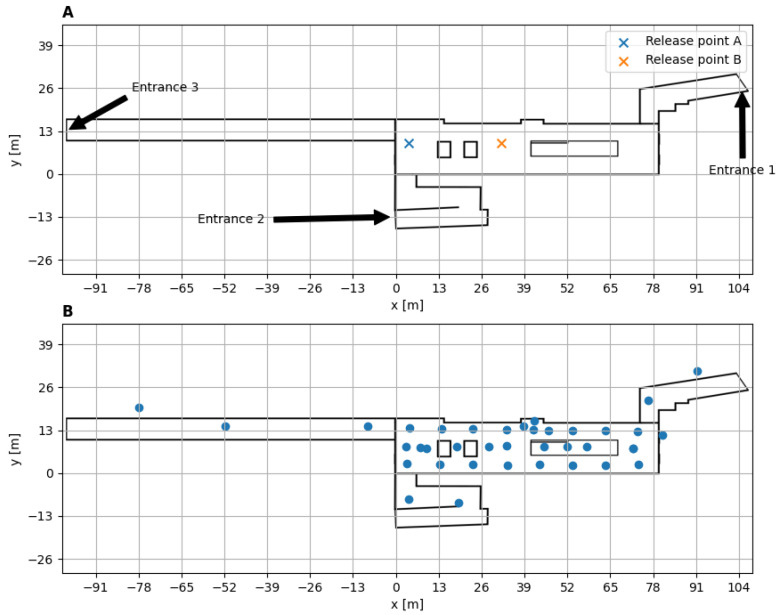
(**A**) The location of release points A and B on the concourse floor. The entrances from the street level are marked. (**B**) The location of NDIR sensors in the concourse at trials A1–A4, and B1–B4. The red cross marks the release point, and the blue circles mark the location of the NDIR.

**Figure 2 sensors-24-05916-f002:**
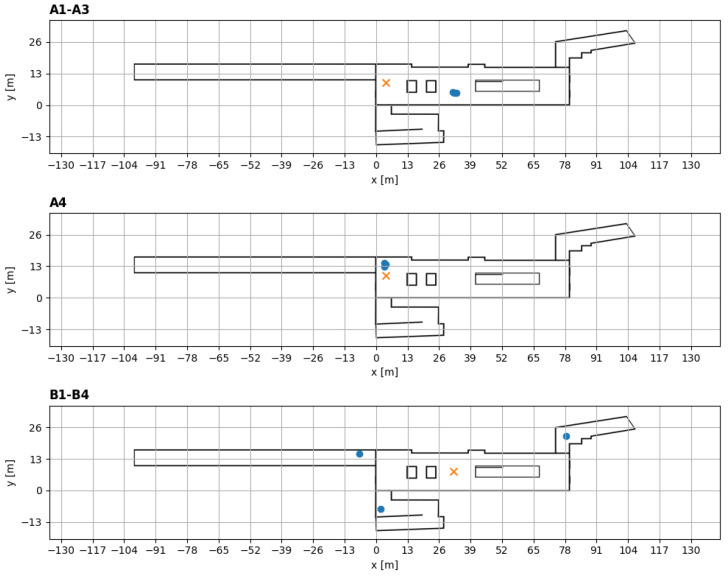
The location of the Kaijo anemometers and the release points in trials A1–A4, and B1–B4. The red cross marks the release point, and the blue circles mark the location of the ultrasonic wind anemometers (KAIJO). Note that, in trials A1–A4, the three Kaijo anemometers were located at the same point but at different heights above ground.

**Figure 3 sensors-24-05916-f003:**
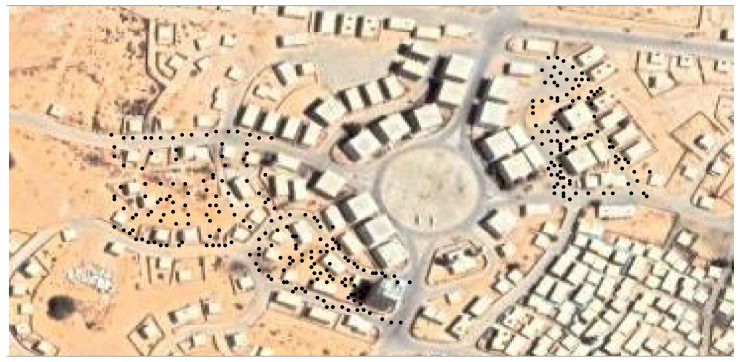
The location of the Petri dishes in the outdoor field campaign. A black dot indicates each Petri dish.

**Figure 4 sensors-24-05916-f004:**
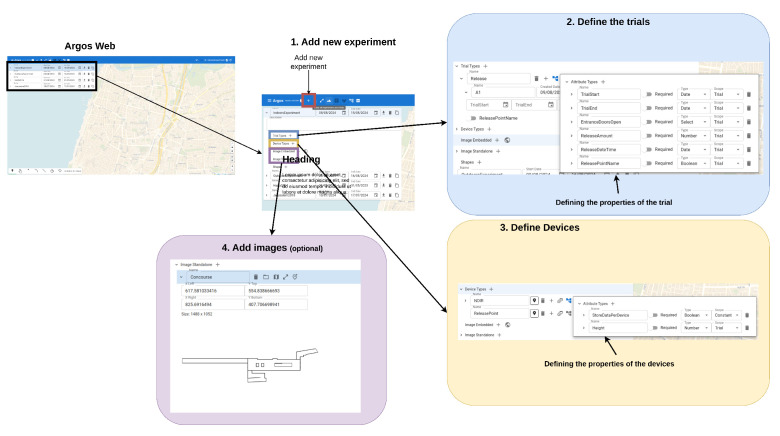
The ArgosWeb interface allows the user to define the field campaign (experiment). That is, the user defines the trials and their properties, as well as the devices and their properties. The deployment of the devices on the map is also determined with this interface (see Figure 5). See the text for a description of the functionality of each box.

**Figure 5 sensors-24-05916-f005:**
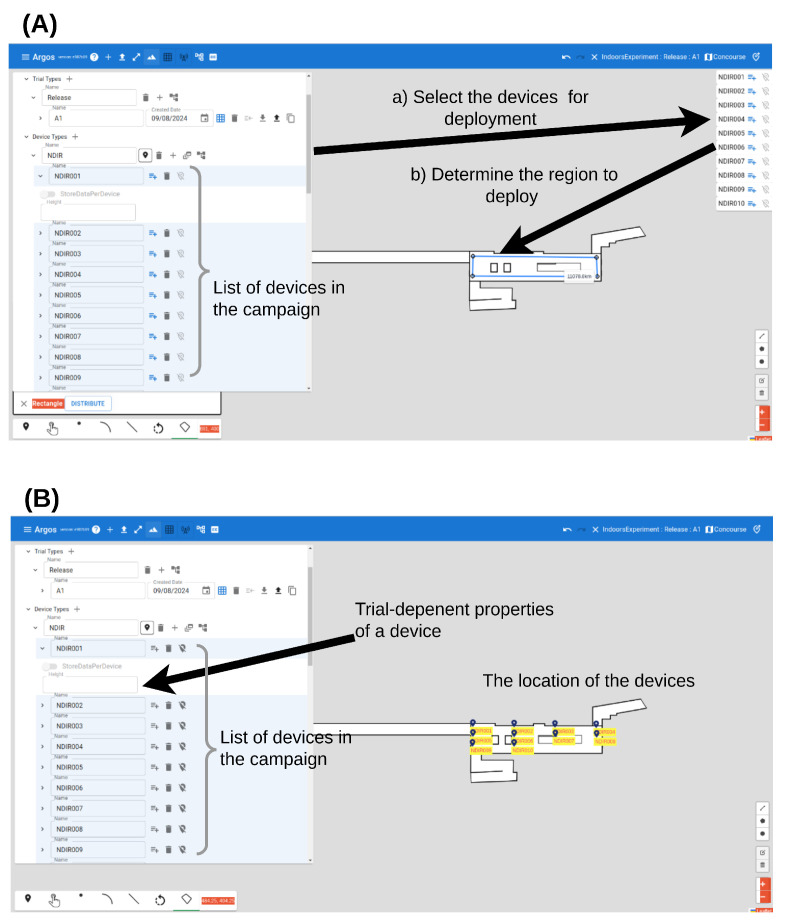
Deploying the device in ArgosWeb interface (**A**). The graphical interface allows the user to select the devices to deploy from the devices that were defined in the campaign (**B**). The devices are deployed in a point, line, arc, or rectangle (the icons on the lower left side of the screen). The interface allows the user to set the trial-dependent properties.

**Figure 6 sensors-24-05916-f006:**
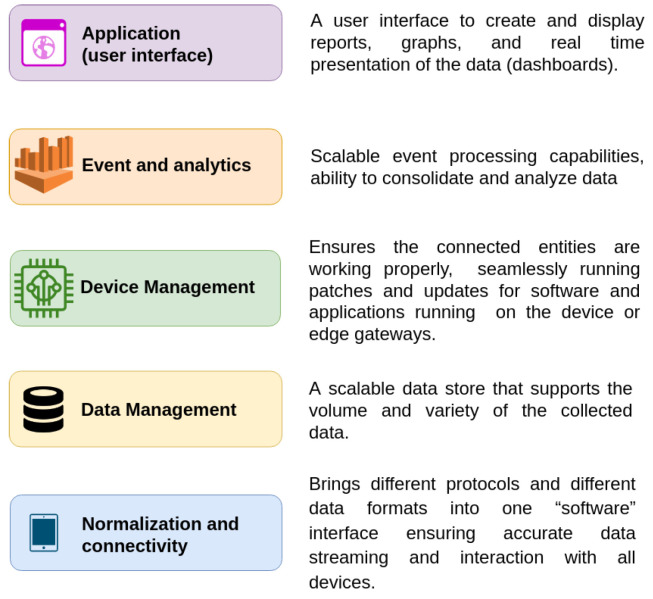
The functional layers of the server domain.

**Figure 7 sensors-24-05916-f007:**
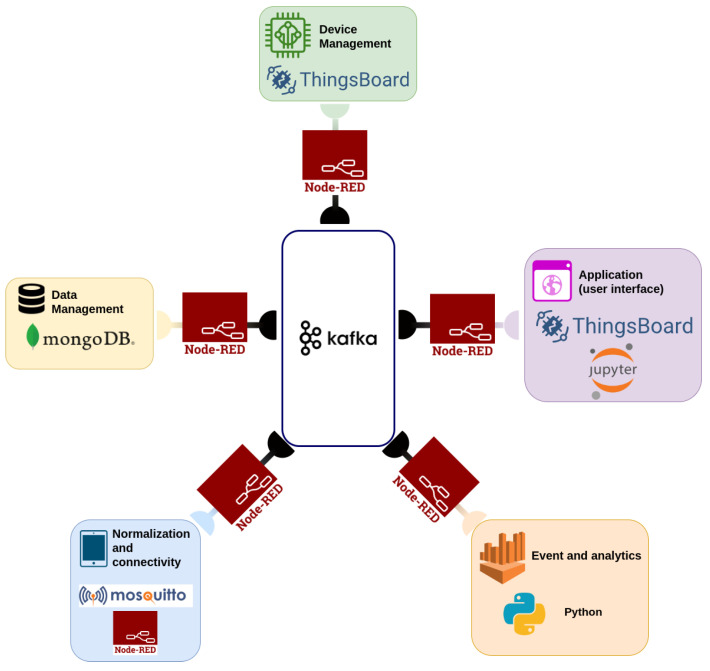
A microservice implementation of the IIoT architecture of Figure 6 using open-source components.

**Figure 8 sensors-24-05916-f008:**
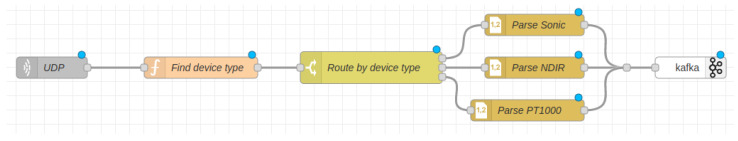
The Node-RED workflow manages real-time data acquisition, parsing and sending the data to a Kafka topic. The parsing procedure is device-specific.

**Figure 9 sensors-24-05916-f009:**
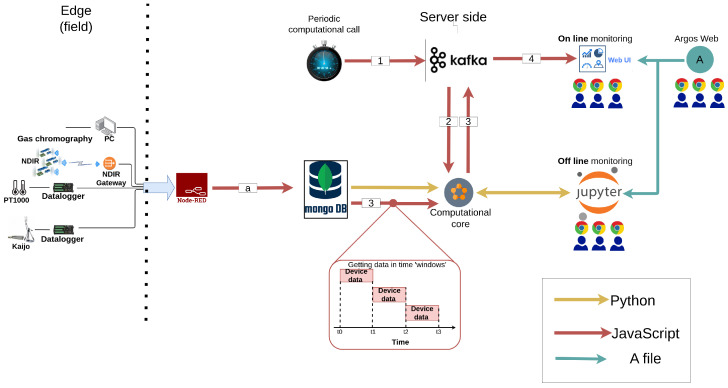
The dataflow in the indoor field campaign.

**Figure 10 sensors-24-05916-f010:**
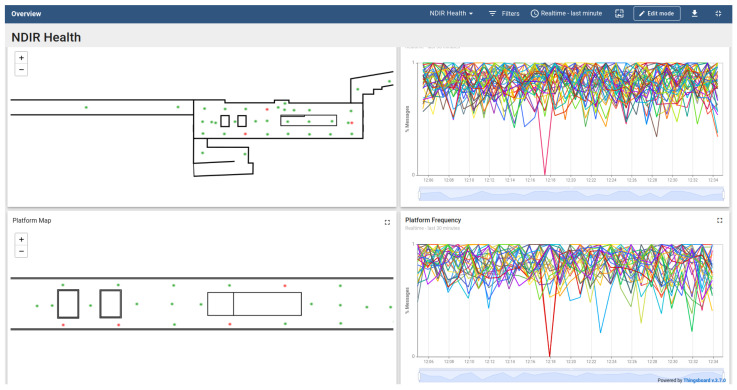
The Thingsboard dashboard that shows the status of the NDIR sensors. (**Left**) The distribution of the NDIR in the concourse and the platform. Green indicates more than 70% messages in the last minute; red indicates less than 30%. (**Right**) The frequency during the last 30 min. Each color indicates a different device.

**Figure 11 sensors-24-05916-f011:**
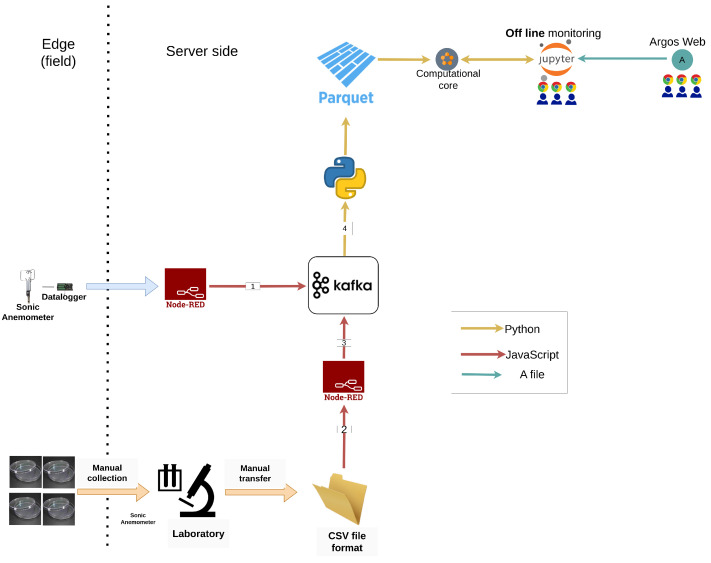
The data flow in the outdoor field campaign.

**Table 1 sensors-24-05916-t001:** Exemplary trials. The listed properties comprise only part of the properties in the field campaign.

Name	Start	End	Release Point	Amount Released (g)	Release Time	Entrance Doors
A1	7-3-2022 9:10	3-7-2022 9:39	A	500	3-7-2022 9:19	open
A2	7-3-2022 11:00	3-7-2022 11:40	A	500	7-3-2022 11:10	open
A3	7-3-2022 11:40	3-7-2022 12:10	A	500	7-3-2022 11:44	closed
A4	7-3-2022 14:15	3-7-2022 14:56	A	500	7-3-2022 14:26	closed
B1	7-3-2022 09:20	3-8-2022 09:50	B	500	7-3-2022 09:30	open
B2	7-3-2022 10:30	3-8-2022 10:55	B	500	7-3-2022 10:35	open
B3	7-3-2022 12:15	3-8-2022 12:49	B	500	7-3-2022 12:19	closed
B4	8-3-2022 14:15	3-8-2022 14:52	B	500	8-3-2022 14:22	closed

## Data Availability

Access to download ArgosWeb and pyArgos will be given upon reasonable request.

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
