# Peer review of "An Industrial Internet-of-Things (IIoT) Open Architecture for Information and Decision Support Systems in Scientific Field Campaigns"

_sensors, 2024, doi:10.3390/s24185916_

Round 1

Reviewer 1 Report

Comments and Suggestions for Authors

The topic of the manuscript is interesting and fits the scope of the Journal. Once read the scarce 9 pages of the manuscript, this reviewer considers that it could be a conference contribution but not a journal paper. It consists on a proposal of a layered architecture using open source software; however, there is no application, implementation descriptions, achieved results, discussion. In fact, there is no novelty in comparison with previous literature, the layers and the software components have been previously reported in various papers already published. For example, the following ones present a very similar architecture applied to different scopes:

A novel integration between service-oriented IoT-based monitoring with open architecture of CNC system monitoring. Int J Adv Manuf Technol 131, 5625–5636 (2024). https://doi.org/10.1007/s00170-022-08675-6

Data acquisition and monitoring system framed in Industrial Internet of Things for PEM hydrogen generators. Internet Things 2023, https://doi.org/10.1016/j.iot.2023.100795

GymHydro: An Innovative Modular Small-Scale Smart Agriculture System for Hydroponic Greenhouses. Electronics 2024, 13, 1366. https://doi.org/10.3390/electronics13071366

IoT based smart ferry system. 2021 J. Phys.: Conf. Ser. 2115 012015 DOI 10.1088/1742-6596/2115/1/012015

Water Level Monitoring System User Interface Using Node-Red. Progress in Engineering Application and Technology Vol. 4 No. 2 (2023) 403–414.

Nevertheless, after a careful revision, the following comments are provided for the enhancement of the manuscript for further submissions.

“Industrial IoT”, apart from “IoT”, could be added as keyword, if the authors agree.

Some abbreviations are not decomposed, being directly used. For example, HTML in line 277 is found.

In the caption of figure 9, “NodeRED” requires a hyphen.

In the introductory section, a common practice consists on placing a final paragraph to briefly describe the structure of the rest of the manuscript. This should be added for a better readability.

Despite the fact that the paper focuses on open source software, the hardware that supports and hosts such software should be mentioned at least in a brief manner. For example, the selected database is MongoDB, where is this database hosted? Where does Node-RED runs?

The reasons to establish 5 layers should be commented. There are similar approaches with 4 or 6 layers in previous literature.

Screenshots of the developed user interface with Thingspeak should be included. The evolution of magnitudes should be commented within such interface.

A block diagram of the architecture depicting the communications and data flow should be added. Figures 2 and 3 are functional but the deployment, including concrete communications and equipment is required.

MQTT is the protocol considered as standard for IoT ecosystems. This feature should be mentioned.

Node-RED is commonly referred to as middleware, so it is desirable to include this term.

Results must be reported and commented, emphasizing the strengths and novelties in comparison with previous literature.

The main limitations of the work and future research guidelines should be included in the last section, at least in a brief manner.

Reviewer 2 Report

Comments and Suggestions for Authors

The authors have presented a software framework based on the Industrial Internet of Things (IIOT) and an implementation using well-established and newly developed open-source components. This framework architecture and components allow developers to customize the software to a campaign’s specific needs while keeping the coding minimum. Using this framework, we developed an Information and decision support system for an indoor dispersion campaign.

1- Just reading this paragraph at the end of the abstract raises the question of whether the authors have actually tested the framework and obtained convincing results. If so, you'd need to back up your statements with the results of these tests.

2- Why test a system in 2022 and wait 2 years later to write the paper? 

3- In Table 1, what do A, B, A1, B1 etc. mean? A good explanation is needed when analyzing and interpreting this table.

4- The authors analyze Table 2 on line 72 of page 2, whereas this table does not exist in the document.

5- After presenting the framework, I expected the authors to study an application case where it has produced convincing results.

6- The discussion and conclusion are two separate sections.

7- The paper needs to go a little deeper in the explanations.

Comments on the Quality of English Language

No comment. 

Round 2

Reviewer 1 Report

Comments and Suggestions for Authors

The revised version of the manuscript has been properly improved.

Author Response

We would like to extend our sincere gratitude to the reviewer for their time and effort in reviewing the second version of our manuscript.

Reviewer 2 Report

Comments and Suggestions for Authors

The authors have certainly made enough effort to improve the quality of the paper, but there are still several limitations to be overcome:

1- When revising the paper, it is important to highlight all the changes you have made in a different colour to make it easier for reviewers to follow your changes. Also, indicate in the reply to reviewers the page or line of changes corresponding to each comment.

2- How the last sentence of the abstract has been modified gives the impression that the authors are sometimes clever. The abstract should be further improved because for the moment it remains vague. The conclusion is also inconsistent and needs to be completely redone.

3- In what type of industry (context) is the application of section 4 illustrated? 

4- Please refer to my comments from the previous evaluation phase for further modifications.

5- Paper referencing should be strengthened using peer-reviewed scientific journal/conference articles.

Comments on the Quality of English Language

No comment.

Author Response

We would like to extend our sincere gratitude to the reviewer for their time and effort in reviewing the second version of our manuscript.

We have accepted all your remarks and updated the manuscript accordingly.  Please see the attachment for the detailed response.  

We apologize for the oversight of omitting the manuscript that highlights the differences between the two versions. We have appended it (and the changes to the present version) at the end of the response. 

Round 3

Reviewer 2 Report

Comments and Suggestions for Authors

The authors have made significant improvements to the document and their efforts to interact with reviewers are commendable. The paper still suffers from a few problems to fix :

-the paper already has a full DOI when it is not even accepted yet. How come? 

-The structure of the abstract and, above all, the conclusion are always correct. For example, the conclusion does not highlight the answer to the problem and future directions, which are two key parts.

-The majority of references are not peer-reviewed articles and are sometimes poorly formatted, which can often be added directly in the text or the form of a footnote.

-Every type of industry has an environmental area to manage, and some are more concerned than others, such as the mining or brewing industries and so on. Now, what type of industry or case study have you tested the framework in?

Author Response

We wish to thank the reviewer for his time and effort reviewing this manuscript. 

We have attached a point-by-point response. 
